# Observed bottom warming in the East Siberian Sea driven by the intensified vertical mixing

Xiaoyu Wang[1, 2, 3], Longjiang Mu[3], and Xianyao Chen[1, 2, 3, *]

[1] Frontier Science Center for Deep Ocean Multispheres and Earth System, Ocean University of China, Qingdao, China

[2] Physical Oceanography Laboratory, Ocean University of China, Qingdao, China

[3] Laoshan Laboratory, Qingdao, China

*Correspondence to*: Xianyao Chen (chenxy@ouc.edu.cn)

**Abstract.**

The East Siberian Sea (ESS) features the broadest continental shelf on Earth and contains nearly 80% of the world's subsea

permafrost. A persistent cold bottom layer, with temperatures at freezing point, inhibits the downward transport of heat, thus preventing the thawing of permafrost and subsequent methane release from sediments. However, in early September 2016, we observed an unprecedented warming of over 3ºC at the bottom of the water column, approximately 46 meters deep in the ESS, following a relatively moderate Arctic cyclone. We attribute this notable bottom warming to enhanced wave-induced vertical mixing, which facilitates the well-mixed Arctic marginal seas and allows surface heat to reach the bottom layer. As

sea ice continues to retreat in the Arctic continental shelf, wind-driven waves have longer fetch to grow. Consequently, even moderate cyclones can trigger substantial vertical mixing, a phenomenon not previously documented. Given the accelerated warming of the Arctic and the rapid decline of sea ice, we anticipate that more open water will foster the growth of larger wind-driven waves and intensified vertical mixing, leading to greater heat influx to the bottom layers of Arctic shelves in the future.

## 1 Introduction

The East Siberian Arctic shelf includes the Laptev Sea, the East Siberian Sea, and the Russian side of the Chukchi Sea. It is the broadest ($\sim 2.1 \times 10^6$ km$^2$) and shallowest (mean depth < 50 m) shelf in the world ocean, with nearly 80% of existing subsea permafrost and up to 50% methane flux of the global coastal seas area (Shakhova et al., 2007; 2019). In summer, the seafloor of the East Siberian Arctic shelf is generally covered by cold bottom waters (<-1.0 ºC) with several to

tens of meters thick (Dmitrenko et al., 2010; Bauch & Cherniavskaia, 2018; Wang et al., 2021). This cold layer acts as a barrier to isolate the sea bed from the warmer surface layer and then reduce the methane emissions from sediments by about half (Ferré et al., 2020; Altuna et al., 2021).

However, recent observations show that the near-bottom water over the Laptev Sea mid-shelf has exhibited pronounced warming due to the increasing downward mixing of surface warm water during the retreat of the surface sea ice

since mid-1980s (Janout & Lenn, 2014; Janout et al., 2016). In the later summer of 2007, the wind-induced vertical mixing

caused a dramatic and abrupt near-bottom warming of about 3.0°C on the Laptev Sea mid-shelf (30-m water depth) (Hölemann et al., 2011). The warming impacts on bottom water temperature anomaly remained for up to 3 months after the wind event, leading to the warm bottom waters during the winter (Kraineva et al., 2019). A series of year-round oceanographic moorings at a deeper location (40-m water depth) in the Laptev Sea recorded near-bottom temperatures

reaching 2.0ºC during 2012/2013 (Janout et al., 2016). Turbulent mixing is critical in determining vertical heat transfer in the water column. The strength of vertical mixing on the East Siberian Arctic shelf is dominated by wind forcing, tide-/shear-induced mixing, surface stress from ice motion, and buoyancy loss during ice formation. When the sea ice declines dramatically at the end of the summer melt season, wind forcing becomes dominant (Janout & Lenn, 2014). Previous studies showed that cyclone-induced strong wind can cause a mixing layer deepening of about 5 - 10 m (Long & Perrie, 2012; Peng

et al., 2021). This slight deepening can only cause significant impacts on the thermohaline bottom conditions on the inner shelf (water depth< 30 m).

The East Siberian Sea (ESS) has a much broader shelf than the Laptev Sea. However, for the lack of in-situ observations, few studies focus on the bottom warming and related physical processes in the ESS. In the summer of 2016, the sea ice within the ESS sector decreased significantly, leaving a nearly ice-free shelf by early September (Figure 1). On

September 11th, a uniformly mixed water column on the mid-shelf of the ESS was observed with the bottom water temperature reaching nearly 3.0ºC. It is the first observation of such remarkable vertical mixing event in the ESS during the period with the most intensified upper stratification over the course of a year. This paper reports this extreme mixing event and the related processes. We also present the intensification of the vertical mixing during the sea ice retreat for the past two decades and its thermal impacts on the bottom layer in the ESS.

The paper is structured as follows. Section 2 provides a data and methods description. Section 3 presents the process of an extreme vertical mixing event firstly observed in the ESS and its influences on the heat budget of shelf waters. Discussion and conclusions are given in Section 4.

## 2 Data and Methods

### 2.1 In-situ observations

The cruise (LA77), using the vessel R/V Akademik M.A. Lavrentyev, carried out two hydrographic sections across the ESS. The cross-shelf observations started at station LA77-17 on September 2, and ended at station LA77-40 on September 15, 2016 (Figure 1b). Twenty-three temperature-salinity profiles were obtained using an SBE-911 plus Conductivity, Temperature, and Depth (CTD) sensor with temperature and salinity accuracies of 0.001°C and 0.003, respectively. The sensor was calibrated at the Sea-bird facilities in Seattle before and after the cruise, and the CTD casts were processed

according to Sea-bird processing procedures. Nine temperature-salinity profiles were obtained using expendable CTD (XCTD) with temperature and salinity accuracies of 0.02°C and 0.04, respectively. The in-situ CTD and XCTD data are

available at . The wind speed at each station was recorded by a mechanical anemometer mounted at about 10 m high.

## 2.2 Reanalysis product and satellite-derived data

We use the hourly and monthly atmospheric reanalysis products provided by the European Center for Medium-Range Weather Forecasts (ECMWF) (Hersbach et al., 2020). The spatial resolution of meteorological and surface heat flux datasets is $0.25° \times 0.25°$. That of the ocean wave datasets is $0.5° \times 0.5°$.

     We use the Global Ocean Physics Reanalysis (GLORYS12V1) product with an eddy-resolving (1/12° horizontal resolution and 50 vertical levels) resolution to evaluate the warming trends of bottom waters on the shelf bottom

(Lellouche, et al., 2021). The model component is the Nucleus for European Modelling of the Ocean (NEMO) platform driven at surface by ECMWF ERA-Interim then ERA5 reanalysis. Along track altimeter data (Sea Level Anomaly), satellite sea surface temperature, sea ice concentration and in-situ temperature and salinity profiles are jointly assimilated. Moreover, a 3D-VAR scheme provides a correction for the slowly-evolving large-scale biases in temperature and salinity. This reanalysis has good representation of Arctic sea ice concentration and water temperature,

and it is widely used in the analysis of Arctic hydrographic structure changes (Hall et al., 2021; Hudson et al., 2024; Ivanov et al., 2024).

     The sea surface temperature (SST) used in the analysis is from the Optimally Interpolated SST daily product v5.1 with a spatial resolution of $0.25° \times 0.25°$, supported by Remote Sensing Systems sponsored by National Oceanographic Partnership Program (NOPP) and the NASA Earth Science Physical Oceanography Program.

## 2.3 Mixed layer depth, wavelength of surface wave, and ocean heat content

     The mixed layer depth (MLD) is usually between 15-25 m on the Arctic shelves, and its change can be used to demonstrate the regionally diapycnal mixing strength (Peralta-Ferriz and Woodgate, 2015). The method to determine the MLD is according to the maximum depth at which the density is within a certain threshold of the shallowest measured density. Due to complex hydrographic conditions around the Arctic Ocean, there are various thresholds from 0.01 to 0.25

$kg/m^3$ (Timmermans et al., 2012; Peralta-Ferriz and Woodgate, 2015). In this study, we use a threshold value of 0.05 $kg/m^3$ to detect the MLD. Note that the other threshold density values would not make a difference in determining the extreme mixing event.

     To evaluate how an extreme diapycnal-mixing event may impact the vertical heat budget on the central shelf of the ESS, the wavelength of surface wave, and the ocean heat content, are considered. The downward maximum mixing depth

caused by wind-induced surface waves is usually consistent with its wavelength (Qiao et al., 2010); therefore, we can infer the maximum mixing depth of surface waves according to its wavelength, $\lambda$ (meter), which is calculated using equations derived from the dispersion relation of regular waves (Holthuijsen, 2007), as follows:

$$\lambda = \frac{gT^2}{2\pi} \qquad \text{when } d/\lambda \geq 0.5,$$

$$\lambda = \frac{gT^2}{2\pi}\tanh(\frac{2\pi d}{\lambda}) \qquad \text{when } 0.05 \leq d/\lambda < 0.5,$$

$$\lambda = \sqrt{gd} \times T \qquad \text{when } d/\lambda \leq 0.05,$$

where d is the water depth in meters, T the wave period in seconds, and g the acceleration of gravity, $9.8$ m/s$^2$. We compute the critical MLD resulted from turbulence for the wave-induced motion following the parameterization of wave-mixed upper ocean layer suggested by Babanin (2006),

$$\text{MLD}_{cr} = \frac{g}{2\omega^2}\ln(\frac{a_0^2 \cdot \omega}{Re_{cr} \cdot \nu}),$$

where $\omega$ is the wave frequency, $a_0$ half of significant wave height, $Re_{cr}$ the critical Reynolds number (~2300), $\nu$ kinematic viscosity of the ocean water (~$1.3\times10^{-6}$) and g the acceleration of gravity ($9.8$ m/s$^2$).

The ocean heat content, Q (Joule), is used to examine to what extent the surface ocean heat being transported to the bottom layer by an extreme diapycnal mixing, which is determined as

$$Q = \int_{z_1}^{z_2} \rho C_p (\theta - \theta_{fz}) d_z,$$

where $\rho$ is average water density (~1026 kg/m$^3$), $C_p$ seawater heat capacity (~4200 J/(kg·°C)), $\theta$ and $\theta_{fz}$ the potential temperature and freezing point (°C) of each layer, respectively, and $z_1$ and $z_2$ the depth range of each layer. The surface layer is the MLD while the bottom layer is from the lower boundary of MLD to the seafloor.

**2.4 Days with heavy winds and high waves**

Energy induced by heavy winds and high ocean waves are primary factors that enhance the diapycnal mixing in the upper ocean. We determine the days with heavy winds and high wave following three criteria: First, the local (in one grid) hourly wind speed is higher than 13 m/s, and its cumulative duration within one day is more than 6 hours; Second, the grids with the wind speed higher than the criteria will occupy at least 20% of the total area of the studied region (155–174.5°E, 71.5–74.5°N); Third, for the days with high waves, its hourly significant wave height must be higher than 2.4 m.

115

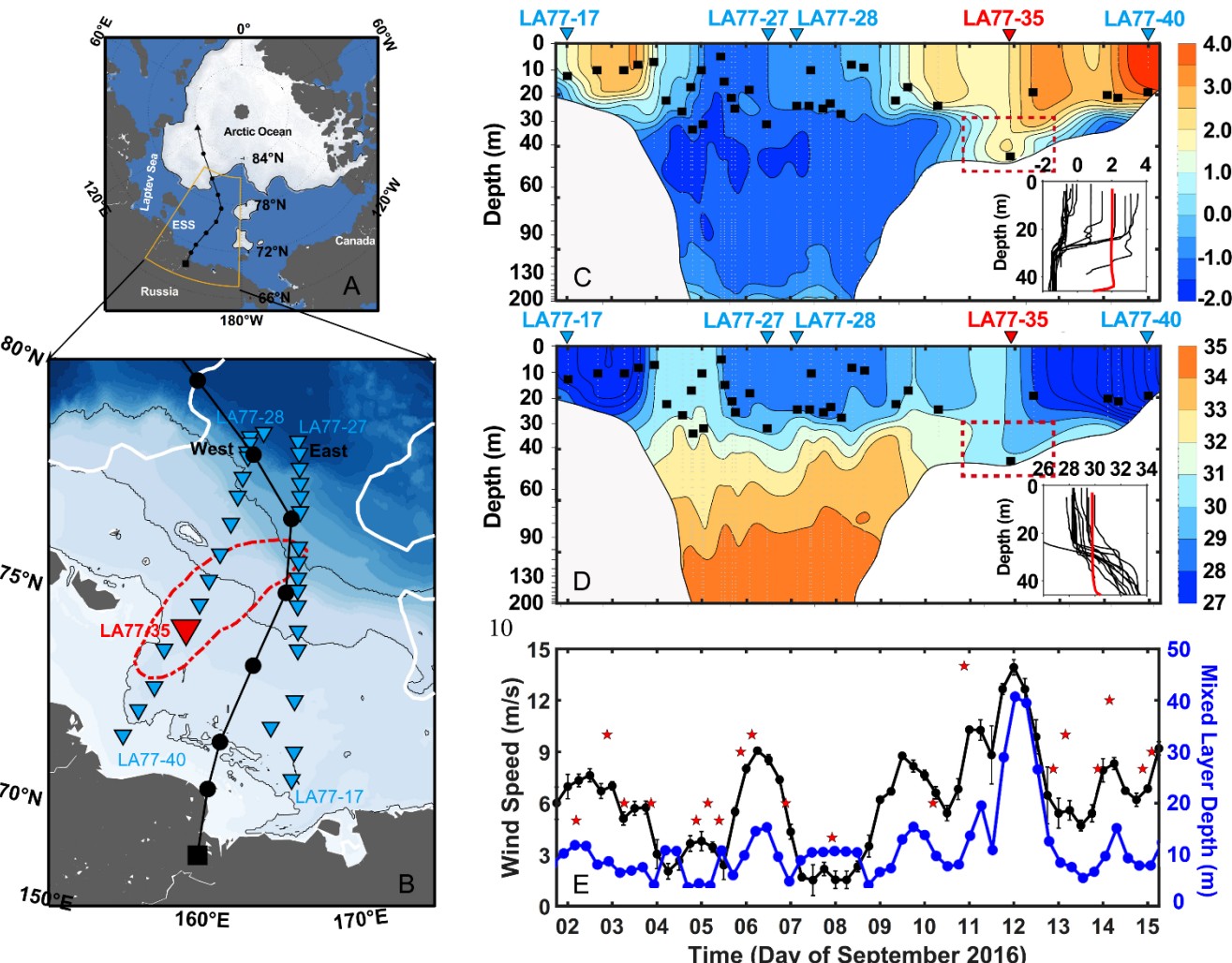

**Figure 1: Observations in the East Siberian Sea (ESS).**

(**a**) The path of the Arctic cyclone from September 9 to 12, 2016. The white area indicates the ocean covered by sea ice. (**b**) Location of stations (blue triangles) in the ESS. The red triangle marks the station LA77-35. Dots show the trajectory of the cyclone center. The red dashed line marks the area with wind speed exceeding 12 m/s. (**c**) The temperature along the section

120    from LA77-17 to LA77-27 and then from LA77-28 to LA77-40. The horizontal coordinate is time. The black square rectangles denote the MLD at each station, and the red triangle marks the location of station LA77-35. The red rectangular box denotes where the significant bottom water warming occurs. The inset figure shows the temperature profiles of all stations, in which, the red line denotes the temperature profile of the station LA77-35. (**d**) Same as (**c**) but for the salinity. (**e**) Variation of wind speeds and $MLD_{cr}$ during the observation period. Black dots denote the 6-hour wind speeds derived from

125    the hourly ERA5 reanalysis data according to the vessel's GPS positions. Blue dots denote the $MLD_{cr}$ induced by wave breaking and wind stress. The red asterisks denote in-situ observations of wind speeds from an onboard anemometer.

## 3 Results

### 3.1 A record deepening of the surface mixed layer in Summer

Observations in September 2016 showed that the water column on the ESS shelf has a two-layer structure: the surface mixed layer and the bottom layer separated by the seasonal pycnocline (Figure 1c and 1d). Temperatures in the surface mixed layer were between 2–5°C, and the salinity was lower than 31.0, while the temperatures in the bottom layer dropped below -1°C, and the salinity ranged between 31.0–34.0. The MLD was only 10–20 m on the mid-shelf, and about 30 m near the slope. However, observation at LA77-35 showed that the mixed layer deepened to the sea floor, resulting in a uniform water column of 45 m thick (Figure 1c and 1d). At the same time, the bottom water temperature increased to about 2°C, above the freezing point. Because station LA77-35 was far from the influences of Atlantic Water and coastal river plumes, we deduce that the unusual bottom warming at LA77-35 was due to intense diapycnal mixing from the surface layer.

From September 9 to 12, 2016, a synoptic-scale Arctic cyclone moved southward across the ESS shelf (Figure 1a) with open water about 1100 km long along its path. When its center moved to the mid-shelf on September 11, the cyclone caused strong winds in the western ESS. Station LA77-35 was right in the center of heaviest wind, where the observed wind speed exceeded 14 m/s on September 11(Figure 1b and 1e). The CTD profile of LA77-35 was obtained on September 12 when the sea condition calmed down. Hence the observed ocean state at LA77-35 represented the final influences of the cyclone on the mid-shelf. The extreme deepening of the mixed layer at LA77-35 was driven by the windy weather condition. The intense diapycnal mixing completely eroded the stratification on the shelf, resulting in a record deepening of the mixed layer in the Pacific sector of the Arctic Ocean during summertime.

### 3.2 Vertical mixing intensified by the growing waves in open water

The sea level pressure at the center of the cyclone was 1012–1024 hPa (Figure 2), indicating it was just a moderate cyclone, compared with the minimum observed sea level pressures during two Arctic cyclones in August 2016 (967 hPa) and August 2008 (976 hPa), respectively (Long & Perrie, 2012; Peng et al., 2021). The resulting maximum wind speed was also lower than the previous two. However, the deepened mixed layer observed in September 2016 in the ESS reached about 25 m depth, much larger than the 5–10 m deepening reported by the previous studies (Long & Perrie, 2012; Peng et al., 2021). The main difference between the cyclone we experienced during September 9-12, 2016, and the reported ones was that its path was along the entirely open waters, allowing the growing of the wave and the continuous wind energy input into the ocean.

At 00:00 UTC on September 11, when the cyclone center was still in the Arctic basin, the significant wave height (using wave height for short in the following text) near the continental slope was about 1.5 m (Figure 2a). Half a day later when the cyclone center moved southward onto the outer shelf, the maximum wave height gradually grew to about 2.0 m. When the cyclone landed on September 12, nearly all the wave heights on the western ESS shelf were above 2.0 m. The maximum wave height exceeded 3.5 m, and its location was close to station LA77-35 (Figure 2c).

The turbulent mixing induced by wave breaking is mainly confined within the near-surface zone with the depth scale of wave heights (Soloviev & Lukas, 2003; Sulisz et al., 2015). To examine whether the wave-induced mixing can reach the seafloor of the mid-shelf, we calculated the wavelength change during the cyclone's passing (Figure 2d–f). Before September 11, the wave length in the mid-shelf region is around 20-30 m. However, after the cyclone passed, the wavelength increased to about 70 m, which is long enough to stir the whole water column with 30-60 m depth by enhanced shearing and wave breaking. We confirm this estimation by calculating the $MLD_{cr}$ induced by wave breaking and wind stress following the parameterization given by Babanin (2006). We find that when the wind speed reaches 14 m/s on the western shelf on September 12, the turbulence process caused by wind and wave breaking can deepen the mixed layer to about 42 m (Figure 1e). This estimation is consistent with the MLD of about 45 m at station LA77-35 (Figure 1c and 1d), indicating that the strong wind-driven turbulence (wave breaking and wind stress) triggered by the cyclone is sufficient to generate strong diapycnal mixing on the shelf and the uniform water column.

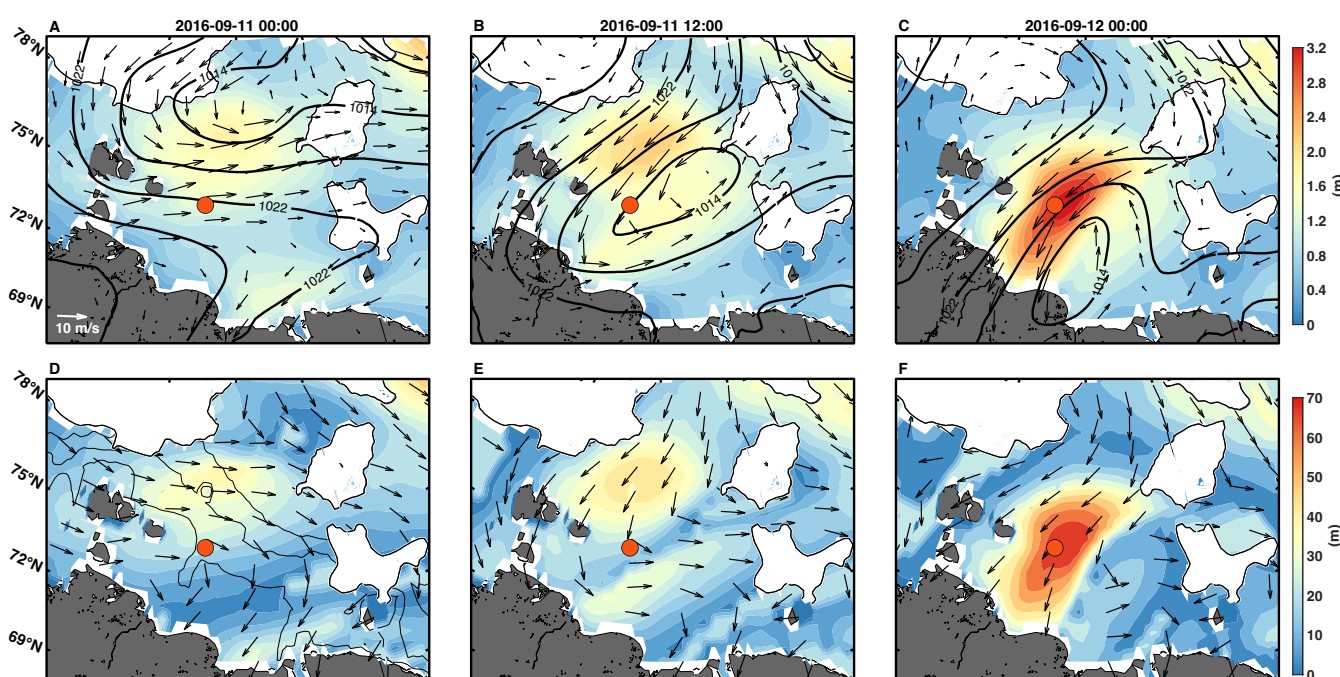

**Figure 2: Distribution of significant wave height (top panel) and the wave length (bottom panel) every 12 hours during September 11-12.**

(**a-c**) The black contours represent sea level pressure in units of hPa. The arrows represent the 10 m wind field. The white area denotes sea ice extent, and the red asterisk marks the location of station LA-77-35. (**d-f**) Arrows indicate the wave direction. Thin black lines in (**d-f**) indicate 30 m, 50 m, and 100 m isobath, respectively.

### 3.3 Heat Transfer to the bottom of mid-shelf by intensified vertical mixing

Observations showed that the MLD in the ESS in early September was only 15 - 25 m, and water temperatures in the bottom layer were below -1°C (Figure 1c). However, the wind-induced diapycnal mixing at station LA77-35 caused a remarking warming of about 3°C near the seafloor of the mid-shelf. This intensified downward mixing process transferred about $1.9 \times 10^8$ J/m$^2$ of heat from the surface to the bottom layer, equivalent to a monthly mean heat flux around 73 W/m$^2$, an order of magnitude higher than the net solar radiation reaching the bottom during summertime.

During September 10-11, the SST at station LA77-35 decreased by about 1.2°C (Figure 3a). This rapid decrease in temperature was associated with the enhanced diapycnal mixing caused by the cyclone, as well as the enhanced surface heat flux from the ocean to the atmosphere. According to the ERA5 Reanalysis data, the net shortwave radiation decreased to about 40 W/m$^2$ in this region (positive value means heat input to the ocean), the sensible and latent heat fluxes at station LA77-35 increased to -40 W/m$^2$ and -50 W/m$^2$, respectively (Figure 3b). The net surface heat flux was about -73 W/m$^2$, which can result in the SST decrease of about 0.15-0.19°C within two days, if assuming a MLD of 15-20 m before the cyclone. The SST at station LA77-35 decreased by more than 1.0°C. The area around the station LA77-35 also experienced same-extent SST cooling based on the satellite observations, suggesting the similar bottom warming induced by intensified vertical mixing as observed at LA77-35.

Note that the reduction of the SST from September 10 to 12 should be more than 2°C under such strong diapycnal mixing of about 45m in depth. But the SST at station LA77-35 only decreased by about 1.2°C. We infer that the additional heat is from the advection of the warming coastal water in the west of the ESS to the mid-shelf (Figure 3e). The heat advection by coastal water only explains about 30-40% of the surface and the mixed layer temperature variability, the enhanced downward mixing of the surface warm water still dominates the bottom warming in this area.

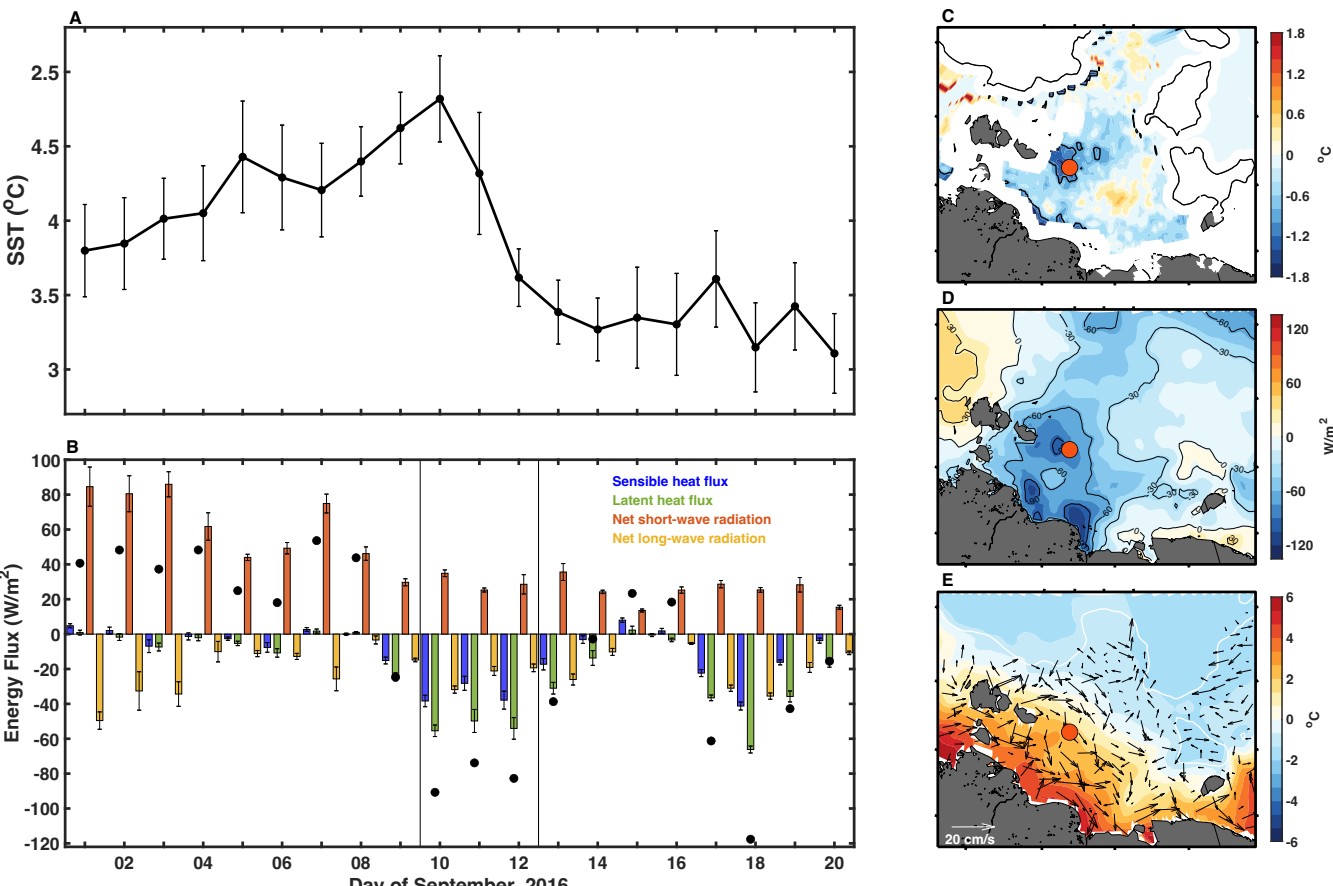

Figure 3: Sea surface temperature (SST), air-sea energy flux, and the surface geostrophic current in the ESS.

(**a**) Changes of mean SST in the region around station LA77-35 (74.375–74.875°N,157.125–157.875°E); (**b**) Same as (**a**) but for the mean air-sea energy fluxes. Positive value means heat input into the ocean. Black dots denote the net surface energy flux. (**c**) Difference of the SST between September 10 and 12. The red dot marks the location of station LA77-35. (**d**) Distribution of mean surface energy balance between September 10 to 12. (**e**) Same as (**d**) but for the mean surface geostrophic current and SST during the observation period.

### 3.4 Intensification of the wind-induced mixing on the ESS shelf

We further investigate the long-term trend of wind-induced mixing in the ESS during the recent decade. The key parameters include the sea ice extent, the day with heavy winds and high waves, the MLD, and the bottom water temperature (Figure 4). Observational and simulated results indicate an intensification of Arctic cyclone activity over the last seven decades, especially in the summer season (Akperov et al. 2019; Karwat et al., 2021; Zhang et al. 2023). The summer sea ice extent became lower than $0.5\times10^6$ km$^2$ since 2007, and remained at similar level since then (Figure 4a). Simultaneously, the

number of days with heavy winds and high waves gradually increased (Figure 4b). All of the top five years with the most prevailing windy processes occur in recent decade. Especially, in September 2016 and 2017, heavy winds and high waves prevailed in the ESS for more than a week (Figure 4b). As a result, the mean MLD in September exhibits significant deepening trend from about 11 m to nearly 15 m during last two decades (Figure 4c), and the bottom temperature reaches nearly 0°C during 2015 and 2016, indicating an increasing influence of wind-induced synoptic processes on vertical mixing (Figure 4d). This intensified vertical mixing may result from a synergistic effect of wave-induced mixing (Mellor 2008; Qiao et al., 2010), wind-induced inertial oscillations (Lenn et al., 2011), the alignment of wind and current directions (Burchard and Rippeth, 2009; Lincoln et al., 2015), or the propagation of shelf waves carrying signals from distant storms (Schulz et al., 2021).

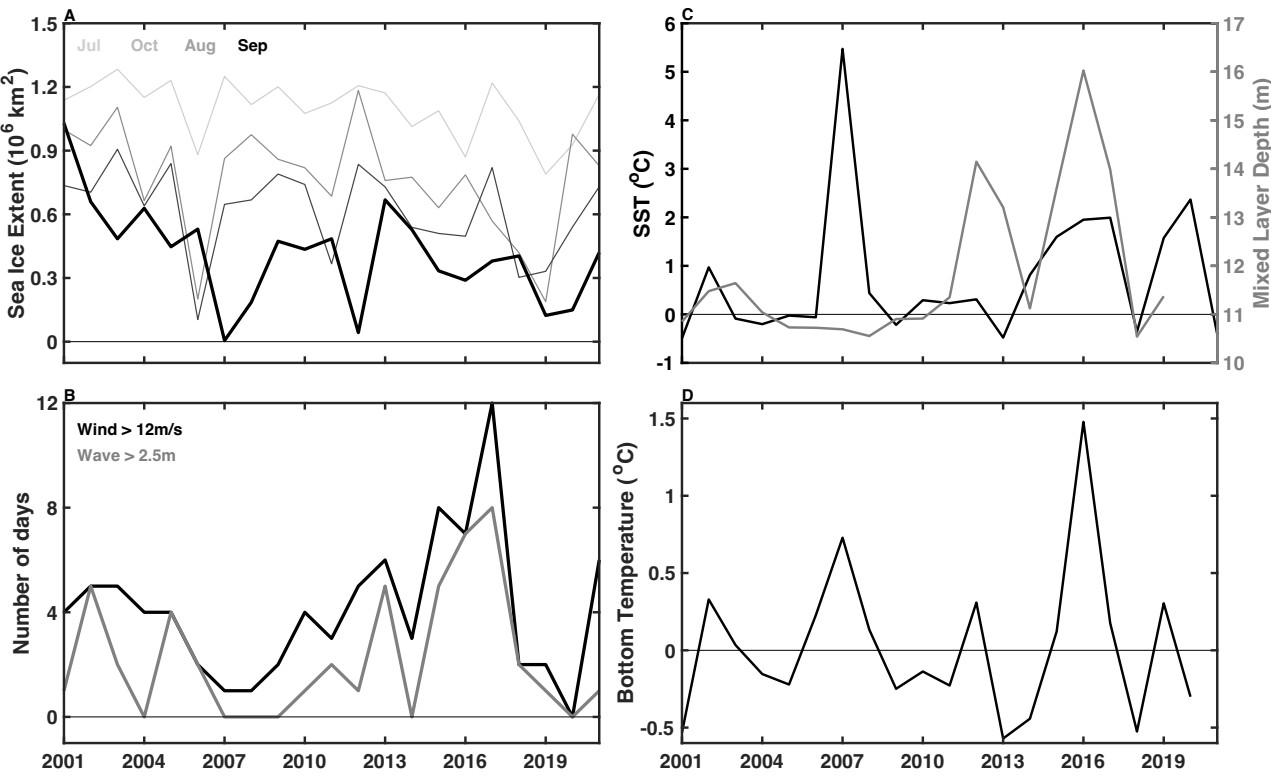

**Figure 4: Variability of the sea ice extent, number of days with heavy winds and high waves, SST, and the bottom water temperature in the ESS during 2001-2021.**

(**a**) Variation of sea ice extent in July, August, September (thick black line), and October in 2016. (**b**) Number of days with heavy winds and high waves in September. (**c**) SST and the MLD in the ESS in summer. (**d**) Same as (**c**) but for the bottom water temperature.

## 4 Discussion and conclusions

In this study, we report an extreme event with a thickened mixed layer to the seafloor with a 25 m depth in the ESS, much deeper than the event reported in previous studies, which deepened the surface mixed layer by 5-10 m (Long & Perrie, 2012; Peng et al., 2021). This recorded-deepening of the mixed layer in the Pacific sector results from an Arctic cyclone moving onto the shelf from open waters with nearly a thousand kilometres from north to south. Over such large open water area, the wave can grow sufficiently and become strong enough to strengthen the vertical mixing of the mid-shelf ocean. The intensified mixing transports a large amount of heat from the upper layer straightly to the seafloor, leading to remarkable warming of about 3°C in the bottom layer. Our observational study provides an important supplement and extension of previous knowledge of wind-induced bottom water warming in the Laptev Sea shelf (Hölemann et al., 2011; Janout et al., 2016; Kraineva et al., 2019).

The Arctic has warmed nearly 2-4 times faster than the global average since 1979 (Serreze & Barry, 2011; Rantanen et al., 2022). At the same time, the summer sea ice covering the Arctic shelves also decreased dramatically and remained at a low level since 2007 (Park et al., 2020; Liu et al., 2021). Lengthening summers and reduced sea ice extent allow more heat into the ocean (Angelopoulos et al., 2020; Dong et al., 2022). With the increasing trends of the open-water area, heavy wind days, and high wave days during summertime since 2012, more vigorous wind-induced mixing and the enhanced bottom warming will happen in the ESS. Now the average bottom water temperature in summer is still below 0°C. However, with the further reduction of sea ice extent and increasing heat content in the mixed layer in the future, the bottom water temperature will undoubtedly become warmer than the freezing point, leading to rapid retreat of bottom permafrost and release of the methane buried in the sediments (Shakhova et al., 2014; James et al., 2016; Sultan et al., 2020; Wild et al., 2022). Quantifying the impact of these processes is beyond the scope of this paper, but it deserves attention in the future.

Observations in the Laptev Sea revealed that the near-bottom warming could remain for at least 2–3 months when the sea ice refreezes again (Janout et al., 2016). This means that the heat stored in the bottom layer during summertime can partly return to the surface layer to melt ice. Unfortunately, the lack of continuous mooring observations prevents us from evaluating how many proportions of heat can reemerge to melt sea ice in the following winter in the ESS. Analysing and quantifying the vertical distribution and transport of ocean heat on the Arctic shelves will improve the understanding of the impacts of the Arctic Ocean warming on regional climate changes. To address this question, long-term mooring systems are urgently needed to monitor the ocean-seabed thermal processes in the ESS.

## Competing interests

The contact author has declared that none of the authors has any competing interests.

## Acknowledgments

This study is supported by the National Key R&D Program of China under Grant 2019YFA0607000,the Chinese Natural Science Foundation under Grants 42276248 and 42176235. The authors would like to thank the European Center for Medium-Range Weather Forecasts for access to the ERA5 reanalysis data, the National Oceanographic Partnership Program (NOPP) and the NASA Earth Science Physical Oceanography Program for access to the Optimally Interpolated SST data, the Copernicus Marine Environment Monitoring Service (CMEMS) for access to the Global Ocean Physics Reanalysis product and the surface geostrophic current data.

The in-situ observation data and the detail introductions are available at https://doi.org/10.5281/zenodo.4507584.

The ERA5 hourly and monthly data are available at https://doi.org/10.24381/cds.adbb2d47 and https://doi.org/10.24381/cds.f17050d7.

The Microwave Optimally Interpolated SST data are produced by Remote Sensing Systems and are available at https://www.remss.com/measurements/sea-surface-temperature/.

The Global Ocean Physics Reanalysis product is available at https://doi.org/10.48670/moi-00021, and the surface geostrophic current data are available at https://doi.org/10.48670/moi-00148.

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
