# Peer review of "Observed bottom warming in the East Siberian Sea driven by the intensified vertical mixing"

_EGUsphere, 2024_

## Referee Comment (RC2)

**Review of the manuscript: "Observed bottom warming in the East Siberian Sea driven by the intensified vertical mixing."**

This manuscript argues that intense wind-generated cyclones over the shallow shelf (less than 40 m deep) in the East Siberian Sea mix the entire water column and are responsible for warming of more than 3 deg C at the bottom of the ocean, having important implications for the submarine permafrost. The study is based on a combination of reanalysis and in situ observations. The manuscript is well written, and the figures are relevant. However, although of great interest, the manuscript lacks actual proofs that: 1) the warming at bottom is caused by wind-driven mixing. 2) this warming observed at only one station lasts long enough and can be observed on a significant spatial area to have a significant impact on the permafrost. I suggest major revisions for this manuscript. My comments are found below.

**Abstract:** it is impossible by reading the abstract to know what the new findings are and what is 'old' knowledge. Please rephrase by highlighting what is new in this work.

l.11: 'worrying trend': very subjective word that should be avoided in an abstract.
L.12: 'high wave': is it surface or internal waves?
l.13: 'enormous heat downward': please quantify
l.14: 'remarkable warming…': how is the warming at depth observed?
l.15: 'undoubtedly suffer more extreme heatwaves': any evidence of that?

**Introduction:**
l.20: CH4 is methane: please indicate in full letters
l.22: indicate the value of the freezing point temperature.
l. 23: how does the cold layer acts as a barrier to reduce methane emissions?
l.27-28: 'abrupt near-bottom warming': it is not clear for me if it has been shown before (then a reference should be added) or if it is part of the results of this manuscript (then it should be deleted from the introduction). Please clarify.
l.29:'up to 3 months': please add 'after the wind event'.
l.29: 'warm bottom waters': are they still at 3 degrees above the freezing point, or have they cool down? Please clarify
l.31: 'reaching 2 deg C during 2012/2013': please add a reference.
l.34: 'wind forcing, and tides become dominant roles': reference?
l.36: 'cause significant impacts on the thermal condition': is there impact only on the thermal conditions or also on the haline conditions? Please clarify.
l.41: maybe expeditions/campaigns rather than fieldwork.
l.42: 'uniformly mixed and the bottom water temperature reaching nearly 3 deg C': this has already been said earlier in the introduction. And again: is that new results or is that part of a previous study? Please clarify.
l.43-44: 'This paper reports this extreme mixing event and the related processes': Is that new result then? From reading the introduction it felt like this was not new result. The processes linked to the extreme mixing should be better explained and there should be better evidence in the manuscript about these processes. See my comments below.
l.44: 'long-term trend': is it based on mooring time series or on observations?
 Please add a paragraph with the plan of the manuscript.

**Data and methods**
l.49: which year is the expedition?
l.52: add 'respectively' after the temperature and salinity accuracies.

Please add a reference or a link to the CTD data, and some information about the calibration of the CTD.

l.56: 'heat flux': is it surface heat flux in the atmosphere? Please clarify.

l.60: please provide more information about the vertical resolution of the model.

How good is the reanalysis on the shelf? Has it been compared with in situ observations? From my knowledge, there is no assimilation of SST under the sea ice, so it might be worth checking how good (or not) the reanalysis is on the ESS.

l.64: Why is the sea surface temperature not from the reanalysis?

l.71: what does the 'rich vertical structures' mean? Please reformulate.

l.73: 'The wavelength': which wavelength is lambda referring to? Is it the one of the internal waves? Surface waves? Please clarify. In general, I find this section confusing. The authors should help the reader to understand why these notions (mixed layer depth, wavelength and ocean heat content) are being introduced, and how they will be used in the rest of the study.

l.79: What is the unit of Q? Same applies for the wavelength.

l.82: How do you define the bottom layer? The definition of the mixed layer is explained at the beginning of this section but not the definition of the bottom layer.

l.87: how many $km^2$ is 20% of the total area of the studied region?

l.88: where does the wave data come from? Please clarify.

**Results**

Section 3.1 Please reference more frequently to the different panels of the figures.

l.115: Where is station LA77-35?

l.118: 'was driven by the windy weather condition': we don't have proof of that with what is shown now in the manuscript. Why couldn't it be the tidal forcing too? There is only one station showing the warming at depth, it could indeed be because of 'perfect timing' with the location of the wind event, but it could also be caused by tidal forcing in my opinion. More evidence and explanations should be brought up, or the conclusions should be tuned down.

L.125: 'lower than the previous two': please quantify.

l.134-135: Please explain more the theory behind this process.

l.136: what is the 'mid-shelf'? please define

l.137: 'with 30 to 60 m depths Increased to about 70m': I don't understand this sentence. Please clarify.

l.156: 'higher than the net solar radiation': please quantify

l.158: 'enhanced diapycnal mixing': Do you have any proof of that? No turbulence measurements are shown in this manuscript.

l.162: how is the heat loss computed from the net surface heat flux? This should be part of the method section.

l.164: 'with the same cooling rate was about 14000 $km^2$': any evidence of that?

A general comment of section 3.3: the authors argue that 'the area with the significant cooling most likely experienced the same extent bottom warming induced by intensified vertical mixing' (l.164-165). At the same time, there is no sign of warming in the other stations of the section, that are within this area I believe (relatively close to the other stations). How is that possible?

l.185: 'since 2007': which season is considered here?

l.185-186: Have the authors tried to fit a regression line between the sea ice extent and the number of days with heavy winds and high waves? Is that significant?

l.187: 'high waves gradually increased': add a reference to the corresponding figure.

**Discussion and conclusion**

l.202-204: This first sentence seems a bit out of the context, as there is no single mention of methane in the result section. Please reformulate.

l.220-222: Again, this is a strong statement, and there is no evidence of that in the manuscript. Please reformulate and nuance this statement.

l.228-229: I am surprised that the warming could remain for at least 2-3 months when it is not observed in the stations in the surrounding. Please elaborate.

**Figure 1:**
- Panel a: How is the trajectory estimated. Add some names (i.e. ESS; Canada; Groenland)
- Panel c and d: it is hard to know where we are on the section compared to panel b. Please add some station names on the section too. Also: why is the section shallow, then getting deep and then shallow again? This is not clear, I thought it is a cross-shelf section so I would expect it to only gets shallower or deeper.
- What is the x-axis of panel b, c and d
- Add labels to the colorbars in panel c and d
- Caption: a) 'from 9 to 12': of which month? What is the box and what is the line? Be clearer. B) 'Observation station locations': what does that mean? Please reformulate. What is the big red dot in the section?
   c) Help the reader by indicating which section is east and which one is west. 'The horizontal coordinate is time': in days? 'The blue rectangular box': isn't it supposed to be red?

In general, I found this figure interesting and useful for the paper, but it gets complicated ad panel c and d mix eularian and lagrangian processes. It will be useful for this figure but also for the rest of the manuscript to try to disentangle the spatial and the temporal component.

**Figure 2:**
- Why is the colored background different in both figures? 'From September 11 to 12': what is shown from September 11 to 12? Is it then averaged over the time period? Is that from the reanalyses?
- Add the scaling arrow in all the panels and make it white.
- Line 148: it should be only d, there are no black lines in figures e and f.

**Figure 3:**

l.179: Add in section data and methods how the geostrophic currents are estimated.

l.177: 'the red asterisk': shouldn't it be the red dot?

---

## Author Comment (AC1)

**Reply to Reviewers**

We appreciate reviewers for returning their comments promptly, and apologize that we did not reply the comments in time. During last three months, we were in a field work in the Arctic Ocean. Sorry for late response.

We thank reviewers for their helpful and insightful comments. We substantially revised the manuscript, following their advice. Here we present a point-by-point response to the reviewers' comments and suggestions.

**1. Reviewer #1**

*This ms tackles and important topic - the warming of deep water in the shelf seas around the Arctic Ocean and by implication the potential impact on the melting of methane hydrates. The paper essentially proposes that increased open water leads to more momentum transfer to TKE and so mixing via larger storm generated surface waves.*

*Whilst there is good published evidence to support the more open water, more mixing due to storms hypothesis (eg. Polyakov et al, 2020, https://doi.org/10.1029/2020GL089469; Rippeth and Fines, 2022, https://doi.org/10.5670/oceanog.2022.103) different mechanisms are invoked. The authors do not consider these alternatives and so the final step in their argument, ie. more mixing via a more vigorous wave field is not proven as the paper stands.*

*These mechanisms include wind triggered inertial oscillations, eg in the Laptev Sea under under both open water and sea ice conditions (Lenn et al., 2011, https://doi.org/10.1175/2010JPO4425.1). Note as shown in this paper the mixing is intermittent with short term events dominating the downward heat flux (how do these heat fluxes compare to those in line 155). The same wind shear alignment mechanism can also drive the deepening of the surface mixed layer (Lincoln et al., https://doi.org/10.1002/2015JC011382). Also, the development of continental shelf waves trigger by remote storms which drive periods of intense mixing. See (Schulz et al., 2021, http://10.1029/2021GL092988).*

We agree with the Reviewer that these well-documented mechanisms can trigger intense vertical mixing. It includes wave-induced mixing in open water, wind-triggered inertial oscillations, wind-shear alignment, the development of continental shelf wave triggered by remote storms, etc. To distinguish the role of each mechanism on the

observed case requires wind, ocean current meter, ocean turbulence, and *in-situ* continuous water level observations. These observations are very rare in the East Siberian Sea, which hinders the quantifications of the local vertical mixing budget.

However, we would like to emphasize that the first mechanism, i.e., the wave-induced mixing in the open water, is sufficient to explain the bottom warming phenomena in our case. The wave-induced vertical mixing can affect ocean depth on the order of the wave length (Mellor 2008; Qiao et al., 2010). As shown in Section 3, the wavelength at the observed station (the red circle in Figure 2) is longer than 60 m, which can break the bottom boundary layer and transport the heat downward to the bottom.

We include mechanisms the Reviewer suggested in Section 3.4, as follows:
"…, indicating an increasing influence of wind-induced synoptic processes on vertical mixing (Figure 4d). *This intensified vertical mixing may result from a synergistic effect of wave-induced mixing (Mellor 2008; Qiao et al., 2010), wind-induced inertial oscillations (Lenn et al., 2011), the alignment of wind and current directions (Burchard and Rippeth, 2009; Lincoln et al., 2015), or the propagation of shelf waves carrying signals from distant storms (Schulz et al., 2021).*"

For the mechanism of the wind triggered inertial oscillations suggested in Lenn et al. (2011), their measurements were taken under 100% ice cover; therefore, they can disentangle the impact directly from the wind or wave. This is different with our case that the ocean is open and the ocean wave was not suppressed by sea ice.

For the mechanism of the wind-shear alignment, the shear spike is a balance between wind, tide and interfacial/bottom friction. To diagnose this effect requires accurate tidal information in the Arctic, which however, is not well constrained in the barotropic tide model due to sea ice cover.

For the mechanism of the development of continental shelf waves triggered by remote storms, it needs continuous observations at fixed locations to capture periodic signals, which is not available yet in the East Siberian Sea.

Reference.

Burchard, H. & Rippeth, T.P., Generation of bulk shear spikes in shallow stratified tidal seas. *Journal of Physical Oceanography*, 39(4), pp.969-985, 2009.

Lenn, Y.-D., T. P. Rippeth, C. P. Old, S. Bacon, I. Polyakov, V. Ivanov, & J. Hölemann, Intermittent intense turbulent mixing under ice in the Laptev Sea continental shelf, *Journal of Physical Oceanography*, 41, 531-547, 2011.

Lincoln, B. J., T. P. Rippeth, & J. H. Simpson, Surface mixed layer deepening through wind shear alignment in a seasonally stratified shallow sea, *Journal of Geophysical Research*, 121, 6021–6034, doi:10.1002/2015JC011382, 2015.

Mellor, G. L., The three dimensional, current and surface wave equations: a revision. *Journal of Physical Oceanography*, 33, 1978-1989, 2008.

Qiao, F., Yuan, Y., Ezer, T., Xia, C., Yang, Y., Lü, X., & Song, Z., A three-dimensional surface wave–ocean circulation coupled model and its initial testing. *Ocean Dynamics*. 60, 1339–1355, 2010.

Schulz, K., Büttner, S., Rogge, A., Janout, M., Hölemann, J., & Rippeth, T. P., Turbulent mixing and the formation of an intermediate nepheloid layer above the Siberian continental shelf break. *Geophysical Research Letters*, 48, e2021GL09298, https://doi.org/10.1029/2021GL092988, 2021.

*A few minor points:*

*line 26: do you mean "increasing down" mixing of heat? 27-28: a citation is required. Is the 'dramatic and abrupt' warming due to complete mixing of the water column (more likely than diapcynal mixing)?*

It is "increased mixing of heat". Hölemann et al. (2011) observed a rapid increase of the near-bottom temperature (~30 m depth) in the Laptev Sea, and attributed it to the intensified vertical mixing of the water column during storms. Our observations show a similar bottom warming in the East Siberia Sea but to the depth ~45 m. It is due to the much stronger and complete mixing of the water column. We add the citation in the manuscript.

Hölemann, J.A., Kirillov, S., Klagge, T., Novikhin, A., Kassens, H. and Timokhov, L., 2011. Near-bottom water warming in the Laptev Sea in response to atmospheric and sea-ice conditions in 2007. *Polar Research*, 30, 6425.

*32: citation required.*

We add the Reference Janout and Lenn (2014), which suggests that the deepening of the pycnocline on the Arctic shelves results from a series of shear and vertical mixing.

Janout, M.A. and Lenn, Y.D., 2014. Semidiurnal tides on the Laptev Sea shelf with implications for shear and vertical mixing. *Journal of Physical Oceanography,* 44, 202-219.

*41-42: I assume that the: "during this period" are your original observations - this*

*needs to be made clearer. Figure 1e - there is no label on the x-axis to say what the numbers refer to?*

Done. The sentence "*During this period, fieldwork in the ESS found that the whole water column on the mid-shelf was uniformly mixed and the bottom water temperature reaching nearly 3.0 ℃.*" is modified to "*On September 11, 2016, a uniformly mixed water column on the mid-shelf of the East Siberian Sea was observed with the bottom water temperature reaching nearly 3.0 ℃*".

The label of x-axis in Figure 1e is added and the whole Figure is updated following the Reviewers' comments.

*126: "previous studies" needs a citation to direct to the pervious studies?*

Done. "*… by the previous studies (Long & Perrie, 2012; Peng et al., 2021).*"

*155: Heat flux - how does this compare to others on the Arctic continental shelf?*

At present, there are few observations of the downward ocean heat flux across the thermocline on the Arctic shelf. Hölemann et al. (2011) reported a sudden warming of ~3 ℃ near the bottom of the Laptev Sea shelf with a depth of only 30 m in late summer 2007. Based on the change of bottom temperature, it is estimated that the downward heat flux is about 30 W/m$^2$. In our works, we observed a similar warming of bottom water happening at a deeper shelf of 45 m. The monthly mean downward heat flux across the thermocline to the ocean bottom can reach nearly 70 W/m$^2$, nearly same as the net heat flux of the sea surface in Summer.

*Figure 3: I am not clear on what the geostrophic currents are showing?*

Figure 3e shows the mean surface geostrophic currents derived from the sea surface height. The color shading in Figure 3e is the sea surface temperature. The geostrophic current is provided in the AVISO datasets.

**2. Reviewer #2**

*Review of the manuscript: "Observed bottom warming in the East Siberian Sea driven by the intensified vertical mixing."*

*This manuscript argues that intense wind-generated cyclones over the shallow shelf (less than 40 m deep) in the East Siberian Sea mix the entire water column and are responsible for warming of more than 3 deg C at the bottom of the ocean, having important implications for the submarine permafrost. The study is based on a combination of reanalysis and in situ observations. The manuscript is well written, and the figures are relevant.*

Thanks.

*However, although of great interest, the manuscript lacks actual proofs that: 1) the warming at bottom is caused by wind-driven mixing. 2) this warming observed at only one station lasts long enough and can be observed on a significant spatial area to have a significant impact on the permafrost. I suggest major revisions for this manuscript. My comments are found below.*

We improve the manuscript to make these two points solid.

*Abstract: it is impossible by reading the abstract to know what the new findings are and what is 'old' knowledge. Please rephrase by highlighting what is new in this work.*

We re-write the abstract to highlight the new finding from our observations. The main finding is the observed warming of more than 3ºC at the shelf bottom around 46 m depth in the ESS for the first time. We emphasize that this notable bottom warming is due to the intensified wave-induced vertical mixing, especially after a relatively moderate Arctic cyclone. Such strong wave-induced vertical mixing happens because the sea ice retreated and the open water allows the growth of the higher wind-driven wave than before.

*l.11: 'worrying trend': very subjective word that should be avoided in an abstract.*

Done. Thanks.

*L.12: 'high wave': is it surface or internal waves?*

It means surface waves with high wave height. We revise it in the abstract.

*l.13: 'enormous heat downward': please quantify*

We quantify the downward heat in the revised abstract.

*l.14: 'remarkable warming…': how is the warming at depth observed?*

We quantify the bottom warming in the revised abstract.

*l.15: 'undoubtedly suffer more extreme heatwaves': any evidence of that?*

It is an implication. We remove this sentence in the Revision.

The revised Abstract is as follows:

The East Siberian Sea (ESS) features the broadest continental shelf on Earth and contains nearly 80% of the world's subsea permafrost. A persistent cold bottom layer, with temperatures at freezing point, inhibits the upward transport of heat, thus preventing the thawing of permafrost and subsequent methane release from sediments. However, in early September 2016, we observed an unprecedented warming of over 3ºC at the seabed, approximately 46 meters deep in the ESS, following a relatively moderate Arctic cyclone. We attribute this notable bottom warming to enhanced wave-induced vertical mixing, which facilitates uniform stirring of the Arctic marginal seas and allows surface heat to reach the bottom layer. As sea ice continues to retreat in the Arctic continental shelf, wind-driven waves have increased space to develop. Consequently, even moderate cyclones can trigger substantial vertical mixing, a phenomenon not previously documented. Given the accelerated warming of the Arctic and the rapid decline of sea ice, we anticipate that more open water will foster the growth of larger wind-driven waves and intensified vertical mixing, leading to greater heat influx to the bottom layers of Arctic shelves in the future.

*Introduction:*
*l.20: CH4 is methane: please indicate in full letters*

Done.

*l.22: indicate the value of the freezing point temperature.*

Done.

*l. 23: how does the cold layer acts as a barrier to reduce methane emissions?*

Ferré et al. (2020) reported that cold bottom-water conditions could substantially reduce the seepage activity to about 43% reduction of total methane release rates. Besides that, the cold bottom water can maintain a stable two-layer vertical structure which isolating the dissolved methane in the bottom layer form escaping into the sea surface.

*l.27-28: 'abrupt near-bottom warming': it is not clear for me if it has been shown before (then a reference should be added) or if it is part of the results of this manuscript (then it should be deleted from the introduction). Please clarify.*

This is a part of previous work. We add the related reference (Hölemann et al., 2011).

*l.29:'up to 3 months': please add 'after the wind event'.*

Done.

*l.29: 'warm bottom waters': are they still at 3 degrees above the freezing point, or have they cool down? Please clarify*

The warming of about 3.0 ºC degrees above the freezing point (~ -1.6 ºC) is the impacts of extreme and short-term synoptic processes. The in-site moorings show that the bottom temperature will cool down during the following 2-4 days by mixing (Janout et al., 2016). But it can maintain a temperature of 0.3-0.4 ºC warmer than the freezing point (Kraineva et al., 2019). We update this part in the Revision.

*l.31: 'reaching 2 deg C during 2012/2013': please add a reference.*

We add the reference (Janout et al., 2016).

*l.34: 'wind forcing, and tides become dominant roles': reference?*

We add the reference (Janout & Lenn, 2014). The shelf of East Siberian Sea is nearly ice-free at the end of the summer melt season, which means surface stress from ice motion and buoyancy loss during ice formation no longer make contributions during the observational period.

*l.36: 'cause significant impacts on the thermal condition': is there impact only on the*

*thermal conditions or also on the haline conditions? Please clarify.*

Thanks. It should be "thermohaline conditions". We update it in the Revision.

*l.41: maybe expeditions/campaigns rather than fieldwork.*

Done. Thanks.

*l.42: 'uniformly mixed and the bottom water temperature reaching nearly 3 deg C': this has already been said earlier in the introduction. And again: is that new results or is that part of a previous study? Please clarify.*

Sorry for this confusion. In the Introduction, we introduce previous works used to observe an extreme event with 3ºC warming in the bottom of Laptev Sea with about 30 m depth. In the East Siberian Sea, there is no historical record of such higher temperature in the bottom with about 45 m depth. Our observation reported in this manuscript is the first time to capture the extreme bottom warming around 3ºC in the East Siberian Sea. We clarify the difference between the warming of 3ºC in the Laptev Sea before and the warming of 3ºC in the East Siberian Sea in our expedition.

*l.43-44: 'This paper reports this extreme mixing event and the related processes': Is that new result then? From reading the introduction it felt like this was not new result. The processes linked to the extreme mixing should be better explained and there should be better evidence in the manuscript about these processes. See my comments below.*

Please find the reply above to 1.42.

*l.44: 'long-term trend': is it based on mooring time series or on observations?*

Hydrographic observations are very rare in the central East Siberian Sea (water depth >30m). Here, the long-term trend was calculated using reanalysis data.

*Please add a paragraph with the plan of the manuscript.*

Done. We add the following descriptions: "*The paper is organized as follows. Section 2 provides a data and methods description. Section 3 provides the results, including the process of an extreme diapycnal mixing event firstly observed in the East Siberian Sea and its influences on the heat budget of shelf waters. Also, increasing trends of wind-induced mixing in the East Siberian Sea are presented in this section. Discussion and conclusions are given in section 4.*".

*Data and methods*

*l.49: which year is the expedition?*

It is 2016. We add it in the revision.

*l.52: add 'respectively' after the temperature and salinity accuracies.*

Done.

*Please add a reference or a link to the CTD data, and some information about the calibration of the CTD.*

Done.

*l.56: 'heat flux': is it surface heat flux in the atmosphere? Please clarify.*

It is surface heat flux in the atmosphere. We add it in the Revision.

*l.60: please provide more information about the vertical resolution of the model. How good is the reanalysis on the shelf? Has it been compared with in situ observations? From my knowledge, there is no assimilation of SST under the sea ice, so it might be worth checking how good (or not) the reanalysis is on the ESS.*
*l.64: Why is the sea surface temperature not from the reanalysis?*

We use the Global Ocean Physics Reanalysis (GLORYS12V1) product with an eddy-resolving (1/12° horizontal resolution and 50 vertical levels). The SST over the open ocean from satellite has already been assimilated in the reanalysis system. The GLORYS12V1 reanalysis dataset is widely used in the analysis of Arctic hydrographic structure changes (Lellouche, et al., 2021; Hall et al., 2021; Liu et al., 2022; Ivanov et al., 2024), including the Arctic shelves (Hudson et al., 2024).

We add the description and the reference of the reanalysis datasets in the Revision.

References.

Hall, S. B., Subrahmanyam, B., Nyadjro, E. S., and Samuelsen, A. (2021), Surface freshwater fluxes in the arctic and subarctic seas during contrasting years of high and low summer sea ice extent, *Remote Sens.*, 13, 1570, https://doi.org/10.3390/rs13081570.

Hudson, P. A., Martin, A. C. H., Josey, S. A., Marzocchi, A., & Angeloudis (2024), A. Drivers of Laptev Sea interannual variability in salinity and temperature. *Ocean Science*, 20, 341-367. http://doi.org/10.5194/os-20-341-2024.

Ivanov, V. V., Danshina, A. V., Smirnov, A. V., & Filchuk, K. V. (2024), Transformation of the Atlantic

water between Svalbard and Franz Joseph Land in the late winter 2018–2019. *Deep Sea Research Part I: Oceanographic*, 206, 104280. https://doi.org/10.1016/j.dsr.2024.104280.

Lellouche, J.-M., Greiner, E., Romain, B.-B., Gilles, G., Angélique, M., Marie, D., Clément, B., Mathieu, H., Olivier, L. G., Charly, R., Tony, C., Charles-Emmanuel, T., Florent, G., Giovanni, R., Mounir, B., Yann, D., and Pierre-Yves, L. T. (2021) The Copernicus Global 1/12° Oceanic and Sea Ice GLORYS12 Reanalysis, *Front. Earth Sci.*, 9, 698876, https://doi.org/10.3389/feart.2021.698876.

Liu, Y., Wang, J., Han, G., Lin, X., Yang, G., and Ji, Q. (2022), Spatio-temporal analysis of east greenland polar front, *Front. Mar. Sci.*, 9, 943457, https://doi.org/10.3389/fmars.2022.943457.

*l.71: what does the 'rich vertical structures' mean? Please reformulate.*

We rephrase the sentence in the Revision.

*l.73: 'The wavelength': which wavelength is lambda referring to? Is it the one of the internal waves? Surface waves? Please clarify. In general, I find this section confusing. The authors should help the reader to understand why these notions (mixed layer depth, wavelength and ocean heat content) are being introduced, and how they will be used in the rest of the study.*

It is the wavelength of the surface wave. We add the detail introduction of the mixed layer depth, wavelength of the surface wave, and the ocean heat content in the Revision.

*l.79: What is the unit of Q? Same applies for the wavelength.*

The unit of Q is Joule. That of the wavelength is meter. We add it in the Revision.

*l.82: How do you define the bottom layer? The definition of the mixed layer is explained at the beginning of this section but not the definition of the bottom layer.*

The water column on the Arctic shelves exhibits two-layer structure. The surface layer is the mixed layer which is warm and relatively fresh. It is well mixed due to wind energy input. Form the seasonal pycnocline to the seafloor, there usually exists a nearly uniform bottom layer with cold and saline waters. The thickness of this bottom is usually 10-30 meters and, it is also well mixed due to tidal mixing. In our paper, the bottom layer is defined as a layer from the seasonal pycnocline to the seafloor with a range of 10-30 meters thick. We add this definition in the Revision.

*l.87: how many km2 is 20% of the total area of the studied region?*

About 4.4 x $10^4$ km$^2$. We add it in the revision.

*l.88: where does the wave data come from? Please clarify.*

The wave data is from the ERA5 reanalysis product with resolution of 0.5° × 0.5°. We introduce it in the Section of Data and Method.

*Results*

*Section 3.1 Please reference more frequently to the different panels of the figures.*

Thanks. Done.

*l.115: Where is station LA77-35?*

We marked the location of LA77-35 in Figure 1b.

*l.118: 'was driven by the windy weather condition': we don't have proof of that with what is shown now in the manuscript. Why couldn't it be the tidal forcing too? There is only one station showing the warming at depth, it could indeed be because of 'perfect timing' with the location of the wind event, but it could also be caused by tidal forcing in my opinion. More evidence and explanations should be brought up, or the conclusions should be tuned down.*

Reviewer #1 raises the same concerns on the mechanism of intensified vertical mixing. Please referrer the first reply to Reviewer #1.

The tidal mixing does play important role on the vertical temperature variability. However, it will have the same impacts on the other stations during the same period. Therefore, tidal mixing only cannot explain the observed intensified bottom warming.

*L.125: 'lower than the previous two': please quantify.*

Done. "The resulting maximum wind speed reached about 14m/s, which was also lower than the maximum wind speeds reported by the previous two (15m/s and 17m/s, respectively)."

*l.134-135: Please explain more the theory behind this process.*

We add a detail description of wave-induced mixing as follows:

The turbulent mixing induced by wave breaking is mainly confined within the nearsurface zone, which approximately half of the total energy dissipation occurring in the depth of 5m (Gemmrich, 2000; Soloviev & Lukas, 2003). But when the wave has well developed, the disturbance energy due to surface-wave orbital velocities is four orders of magnitude higher than that of the turbulence signal (Soloviev and Lukas, 2003). In this situation, the influence of wave-induced mixing depth can be as large as 25 times the height of significant wave (Terray et al., 1996; Qiao 2004).

*l.136: what is the 'mid-shelf'? please define*

Based on water depths, the shelf region is usually divided into inner-shelf (water depths between 0m and 30m), mid-shelf (between 30m and 100m) and outer-shelf (between 100m and 200m). Here, the mid-shelf is about 30 - 100m region.

*l.137: 'with 30 to 60 m depths Increased to about 70m': I don't understand this sentence. Please clarify.*

We rephrase the sentence as follows: Before September 11, the wave length in the mid-shelf region is around 20-30 m. However, after that, the wavelength increased to about 70 m, which is long enough to stir the whole water column with 30-60 m depth by enhanced shearing and wave breaking.

*l.156: 'higher than the net solar radiation': please quantify*

About 98% of the net solar radiation is absorbed in the upper 30m within the Arctic shelves (Hill, 2008). This means that the net solar radiation penetrating into the bottom layer (usually from 25m to the bottom) is no more than $10W/m^2$ even under a clear sky.

Hill, V. J. (2008). Impacts of chromophoric dissolved organic material on surface ocean heating in the Chukchi Sea. *Journal of Geophysical Research*, 113(C7). https://doi.org/10.1029/2007JC004119

*l.158: 'enhanced diapycnal mixing': Do you have any proof of that? No turbulence measurements are shown in this manuscript.*

We did not have turbulence measurements during this cruise. The process of diapycnal mixing is what we estimate according to the gradual change of the mixed layer depth during observational period and based on the common condition in the East Siberian Sea. Usually, in the East Siberian Sea, there is a two-layer structure on the Arctic

shelves during summer. The surface mixed layer mainly consists of river runoff and sea ice melt water, leading to a strong seasonal pycnocline. The pycnocline prevent further downward diapycnal mixing. At the same time, waters near the seafloor are well mixed by bottom tidal mixing, but this mixing process near the seafloor cannot destroy the seasonal pycnocline. Therefore, only the enhanced diapycnal mixing can erase the pycnocline and cause a uniform water column of nearly 50 meters thick.

*l.162: how is the heat loss computed from the net surface heat flux? This should be part of the method section.*

Done. We add a short description in the method section.

*l.164: 'with the same cooling rate was about 14000 km2': any evidence of that?*

The area of 14000 $km^2$ is an approximate estimation of the region near LA77-35, whose sea surface temperature decreased more than 1ºC. Based on observations, the SST can decrease at most 0.2ºC within two days under the surface cooling but without cold bottom waters being mixed. However, when a cyclone passed across the sea, it can bring a cold sea surface along its pathway by mixing subsurface cold water. The observed SST showed that there existed a cooling of ~1.2 ºC around Station LA77-35. This means that the shelf regions around LA77-35 with a similar water depth and similar temperature cooling are very likely suffered significant mixing. Here we use this property to estimate how much area of the shelf occurs intensified mixing like LA77-35. We add the description on how to estimate the cooling area in the Revision.

*A general comment of section 3.3: the authors argue that 'the area with the significant cooling most likely experienced the same extent bottom warming induced by intensified vertical mixing' (l.164-165). At the same time, there is no sign of warming in the other stations of the section, that are within this area I believe (relatively close to the other stations). How is that possible?*

Stations LA77-34, LA77-35, and LA77-36 locates within the significant cooling region. But only the LA77-35 and LA77-36 were observed after the extreme mixing event. The LA77-35 (red lines in Figure S1) was observed as soon as the high waves calm down; therefore, its vertical structure showed an ideal well-mixed water column. The LA77-36 (blue lines in Figure S1) was observed one day later after the extreme mixing event and also showed an obvious warming signal with a thicker and warmer mixed layer,

which is consistent with that observed at LA77-35. But the water depth at LA77-36 is only 37 m. Therefore, the bottom layer at the LA77-36 shows slight cooling, especially after more than one day of the intensified mixing.

[Figure]

Figure S1. (a)Temperature, (b)salinity and (c)potential density profiles of Stations LA77-29, …, LA77-41 in the western East Siberian Sea.

*l.185: 'since 2007': which season is considered here?*

It is summer season.

*l.185-186: Have the authors tried to fit a regression line between the sea ice extent and the number of days with heavy winds and high waves? Is that significant?*

The area with high wave (>1.4m) while duration (>12h) is negatively correlated well with sea ice extent (*R* is -0.4 with significance level 0.9), as shown below.

[Figure]

*l.187: 'high waves gradually increased': add a reference to the corresponding figure.*

Done.

*Discussion and conclusion*

*l.202-204: This first sentence seems a bit out of the context, as there is no single mention of methane in the result section. Please reformulate.*

Thanks. We remove this sentence in the Revision.

*l.220-222: Again, this is a strong statement, and there is no evidence of that in the manuscript. Please reformulate and nuance this statement.*

Thanks. We remove this sentence in the Revision.

*l.228-229: I am surprised that the warming could remain for at least 2-3 months when it is not observed in the stations in the surrounding. Please elaborate.*

Based on mooring observations in the Laptev Sea, Janout et al. (2016) showed that the warming influence, i.e. ocean heat stored in the bottom layer, could remain for at least 2-3 months until it is being ventilated again by winter convection. Janout et al. (2016) gave a schematic diagram to show how the heat in the surface layer is involved into the bottom layer and released into the atmosphere until winter, as follows.

[Figure]

**Figure 2.** An idealized year on the Laptev Sea shelf, summarizing dominant processes and vertical water column structure. Colors indicate density (dominated by salinity), where lighter colors are less dense. The seasonal progression of the pycnocline is based on the layer of maximum shear from ADCP records. The red shading indicates the spatiotemporal distribution of surface-warmed water. Water temperatures are indicated as "near freezing ($T \approx T_{fr}$)", "above freezing ($T > T_{fr}$)", and "significantly above freezing ($T \gg T_{fr}$)". The color bar indicates air-sea heat fluxes. "Episodic advection" events were observed in 2010 and 2013 (Figure 1 and *Janout et al.* [2013]). Advection of fresher surface waters from shallower regions counters brine rejection during early winter, and enables the preservation of warmer near-bottom waters.

*Figure 1:*

*- Panel a: How is the trajectory estimated. Add some names (i.e. ESS; Canada; Groenland)*

The trajectory is determined as the center of the low sea level pressure every 6 hours.

*- Panel c and d: it is hard to know where we are on the section compared to panel b. Please add some station names on the section too. Also: why is the section shallow, then getting deep and then shallow again? This is not clear, I thought it is a cross-shelf section so I would expect it to only gets shallower or deeper.*

Figure 1C shows the temperature of each station from the station LA77-17 to LA77-27 along the east cruise line, and then from the station LA77-28 to LA77-40 along the west cruise line. Therefore, the depth along the section is from shallow to deep, and getting shallow again. We add the station information in Figure 1C to clarify the meaning of the temperature and salinity profiles.

*- What is the x-axis of panel b, c and d*

It is time (day of September 2016). Added in the Revision.

*- Add labels to the colorbars in panel c and d*

Done.

*- Caption: a) 'from 9 to 12': of which month? What is the box and what is the line? Be clearer.*

Done. Sorry.

*B) 'Observation station locations': what does that mean? Please reformulate. What is the big red dot in the section?*

We rephrase it as "Location of stations". The big red dot is not discussed. We remove it in the Revision.

*c) Help the reader by indicating which section is east and which one is west. 'The horizontal coordinate is time': in days? 'The blue rectangular box': isn't it supposed to be red?*

Done.

*In general, I found this figure interesting and useful for the paper, but it gets complicated ad panel c and d mix eularian and lagrangian processes. It will be useful for this figure but also for the rest of the manuscript to try to disentangle the spatial and the temporal component.*

The horizontal axis of Figure 1C, D, E is the day of September, 2016. Therefore, it shows the temporal variability of temperature, salinity, and wind along the field works when we did the observations. Figure 1B shows the order of the stations and its spatial location. Showing Figure 1B, C,D,E together is helpful for readers to locate the stations and to find the corresponding temperature, salinity, and wind variability. We try to separate temporal and spatial information into two figures, but find that it is hard to compare them back and forth.

*Figure 2:*
*- Why is the colored background different in both figures?*

The colored background in top panels in Figure 2 is significant wave height, and that in the bottom panels is the wave length. We rewrite the caption to make it clear.

*'From September 11 to 12': what is shown from September 11 to 12? Is it then averaged over the time period? Is that from the reanalyses?*

It is the averaged field within each 12 hours from September 11 to 12. The data is from ERA5 datasets. We give the introduction in the Section of Data and Method in the Revision.

*- Add the scaling arrow in all the panels and make it white.*

Done.

*- Line 148: it should be only d, there are no black lines in figures e and f.*

Done. Thanks.

*Figure 3:*
*l.179: Add in section data and methods how the geostrophic currents are estimated.*

Done.

*l.177: 'the red asterisk': shouldn't it be the red dot?*

Done. Thanks.

---

## Author Response (AR2)

**Reply to Reviewers**

We thank reviewers for their insightful comments, which really help us improve the scientific and presentation qualities of the manuscript. We substantially revised the manuscript, following their advice. Here we present a point-by-point response to the reviewers' comments and suggestions.

**1. Reviewer #1**

*The paper presents observation of temperature structure across the East Siberian Sea with one particularly interesting profile showing the water column to be well mixed thus exposing the seabed to warmer water mixed down from the surface.*

*My main criticism of the paper as it stands is that the evidence used to support the claim that the water column is completely mixed is weak – ie the wavelength matches the water depth at the location of the completely mixed water column.*

*I would suggest to make this work more publishable additional evidence should be sort. For example, an using an energy argument, such as the calculation of the different components of potential energy anomaly (see for example the analysis in Rippeth et al, 2001, Journal of Physical Oceanography, 31(8)). In the way the relative contributions of tidal, direct wind, waves etc can be calculated to support the idea that the vertical mixing is due to waves.*

Thank you for your constructive suggestions. Answering this question makes our result more solid. We calculate the maximum mixed layer depth due to wind- and wave-energy dissipations, using the parameterization scheme of wave-mixed upper ocean layer developed by Babanin (2006). The result is consistent with our previous estimations. We add the new result in Figure 1 and provide more explanation and references.

The parameterization scheme developed by Babanin (2006) is evaluated by the observations given in US. Naval Research Laboratory (Young et al. 2011; Liu et al. 2019). Based on this parameterization, we find that with the wind speed reaching 14 m/s on the western shelf on September 12, the significant wave height at station LA77-35 can increase to more than 3.5 m, and the turbulence process caused by wind and

wave breaking could deepen the mixed layer to about 42 m. This estimation is consistent with the mixed layer thickness of about 45 m observed by us, indicating that the strong wind-driven turbulence (wave breaking and wind stress) brought by the cyclone is sufficient to generate strong diapycnal mixing on the shelf and the uniform water column.

According to Rippeth et al. (2001), the rate of change of PAE due to tides is 1.4 x $10^{-4}$ W/m$^2$. Tidal mixing is also strong. However, based on Janout and Lenn (2014), the tides are enhanced under sea ice. The tidal mixing is doubled with the presence of the sea ice. Nevertheless, the bottom thermal structure can even maintain when the shelf sea is covered by sea ice. This indicates that the bottom-reached mixing we observed is not due to tidal mixing, but the enhanced wave-induced mixing.

1.  Babanin, A. V. (2006), On a wave-induced turbulence and a wave-mixed upper ocean layer. Geophys. Res. Lett. 33, https://doi.org/10.1029/2006GL027308.
2.  Janout, M. A., & Lenn, Y. (2014), Semidiurnal Tides on the Laptev Sea Shelf with Implications for Shear and Vertical Mixing. Journal of Physical Oceanography, 44(1), 202-219.
3.  Liu, Q., Rogers, W. E., Babanin, A. V., Young, I. R., Romero, L., Zieger, S., Qiao, F., and Guan, C. (2019), Observation-Based Source Terms in the Third-Generation Wave Model WAVEWATCH III: Updates and Verification. Journal of Physical Oceanography, 49(2), 489-517.
4.  Rippeth, T. P., Fisher, N. R., & Simpson, J. H. (2001), The Cycle of Turbulent Dissipation in the Presence of Tidal Straining. Journal of Physical Oceanography, 31, 2458-2471.
5.  Young, I. R., Zieger, S., and Babanin, A. V. (2011), Global trends in wind speed and wave height. Science, 332(6028), 451-455.

*Other points:*

*Line 9: Surely downward not upward?*

Thanks. It is "downward".

*11: seabed – are we talking about the water temperature at the bottom of the water column, or the temperature of the seabed? Quite an important distinction.*

It is the water temperature at the bottom of the water column. We clarify it in the Revision.

*13: Uniform stirring, normally referred to as a well mixed water column*

Yes. We rephrase the sentence as follows:

"*We attribute this notable bottom warming to enhanced wave-induced vertical mixing, which facilitates the well-mixed Arctic marginal seas and allows surface heat to reach the bottom layer.*"

*14: Increased space to develop. Do you mean a longer wave fetch?*

It is longer wave fetch. The word "increased" in the sentence is misleading. We correct it in the Revision, as follows:

"*As sea ice continues to retreat in the Arctic continental shelf, wind-driven waves have longer fetch to grow*"

*22: Does the 50% refer to a methane flux or is it the amount of methane stored in the sea bed?*

It means 50% of the methane flux for the global total coastal seas area, as given in Shakhova et al. (2007), as follows:

"*This area, representing only ~13% of the global area of the coastal seas (27 × 10$^6$ km$^2$), would generate up to 50% of the 1 Tg CH$_4$ yr$^{-1}$ flux given in Cynar and Yayanos (1993) for the total coastal seas area.*"

We add this reference in the Revision.

Shakhova, N., & I. Semiletov, Methane release and coastal environment in the East Siberian Arctic shelf, *Journal of Marine Systems*, 66, 227-243, 2007.

*25: Barrier? You mean the cold water isolates the sea bed from the warmer surface layer.*

Yes. We clarify it in the Revision as follows:

" This cold layer acts as a barrier to isolate the sea bed from the warmer surface layer and then reduce the methane emissions from sediments by about half "

*27: Downward heat transport. Do you mean mixing down of heat? Try and be consistent.*

It is the downward mixing of surface warm water. We use this description throughout the Revision.

*54: I don't understand why the tides only become dominant under ice free conditions? You need to explain this.*

Thank you for the careful comments on this issue. Following Janout and Lenn (2014), the tides are enhanced under sea ice. During the sea ice retreat, the tides become relatively weak. We remove "the tides" in the Revision.

This also helps us understand the relative roles between wave-induced mixing and tidal mixing. The tidal mixing is doubled with the presence of the sea ice. Nevertheless, the bottom thermal structure can even maintain. This indicates that the bottom-reached mixing we observed is not due to tidal mixing, but the enhanced wave-induced mixing.

*63: "long term" is generally taken to mean over a number of years, you are only considering a season?*

In Line 63 and Line 261 in previous tracked-changes version, we did use "long-term" to describe the trend over a decade.

Line 63 "We also present the long-term vertical mixing intensification during the sea ice retreat and its thermal impacts on the bottom layer in the ESS in the recent decade."

Line 261 "We further investigate the long-term trend of wind-induced mixing in the ESS during the recent decade."

We remove the word "long-term" in the Revision and rewrite sentences with the specified period for clear meaning.

*63: "Intensification" do you mean the mixing is stronger?*

"Intensification" means the stronger mixing. We clarify it in the Revision.

*240: calculation showing the SST doesn't drop enough and so advection must be moving heat? Is this change consistent with the lateral gradients you have measured in the CTD survey?*

Our CTD observations are consistent with the SST data from satellite.

*284: Discussion – I would start by summarising the results before discussing the conseuqneces?*

Following your suggestions, we revise the section of Conclusions and Discussion.

*292: "Straightly" – do you mean mixed down? Please be consistent with language.*

Thank you for pointing this out. It means "mixed down the heat". We correct it in the Revision.

We carefully check the words used in the manuscript to make it consistent and smoothly.

*296: Is the 14,000 km2 the area of the sea bed? It is not clear what you mean.*

The 14,000 $km^2$ is the sea surface area with SST cooling, derived from the SST datasets. While this number is not used for comparison, the physical meaning is not clear, we remove it in the Revision.

*299: Is it increasing solar radiation, or increasing exposure to solar heating due to sea ice retreat?*

It is increasing exposure to solar heating due to sea ice retreat. We make it clear in the Revision.

*301: You need a reference to support your statement that there are more storms in the summer.*

The following three references are added in the Revision.

Akperov, M., A. Rinke, I. I. Mokhov, and et al. (2019), Future projections of cyclone activity in the Arctic for the 21st century from regional climate models (Arctic-CORDEX), *Global and Planetary Change*, 182, 103005

Karwat, A., Franzke, C. L. E., & Blender, R. (2022), Long-term trends of Northern Hemispheric winter cyclones in the extended ERA5 reanalysis. *Journal of Geophysical Research: Atmospheres*, 127, e2022JD036952. https://doi. org/10.1029/2022JD036952

Zhang, X. D., H. Tang, J. Zhang, J. E. Walsh, E. L. Roesler, B. Hillman, T. J. Ballinger & W. Weijer, 2023, Arctic cyclones have become more intense and longer-lived over the past seven decades, *Communications Earth & Environemnt*, 4, 348, https://doi.org/10.1038/s43247-023-01003-0

**2. Reviewer #2**

*The authors answered to all my comments, and I think it helps improving the quality of the manuscript. I don't have any more major comments, and I think that the manuscript is suited for publication.*

*A few typos that I noticed in the manuscript:*

*l.175: 108 J/m2 should be 108 J/m2*

*l.181: 'The net surface heat flux surface': delete surface*

*l.190: 'still dominants' should be 'still dominates'*

Thank you very much for your insightful comments to help us improve the quality of the manuscript.

Thank you for pointing out these typos. Sorry. We corrected them in the Revision.